# A META analysis and systematic review of the effects of exercise interventions on middle-aged and elderly patients with depression

Zheng Zhang[1]*, Jia-Yun Liu[2], Kai-Tuo Zhu[1], Gui-Quan Huo[1]

1 College of Sports Science, Kyonggi University, Suwon, Gyeonggi Province, South Korea, 2 Wuhan Institute of Martial Arts, Wuhan Sports University, Wuhan, Hubei Province, China

* 735142290@qq.com

**Data Availability Statement:** As this study is a meta-analysis, the data used are exclusively from publicly available literature. These literatures have been cited in detail in the paper. We have ensured

## Abstract

### Objective

This study sought to conduct a meticulous systematic review, delving into the efficacy of exercise interventions on depressive symptoms among middle-aged and older adults. Additionally, we aimed to scrutinize the nuanced influence of various intervention factors on the intricate relationship between exercise and depressive symptoms within this demographic.

### Methods

Our investigation involved an extensive search across multiple databases, including PubMed, Embase, Web of Science, and the Cochrane Library, spanning from the inception of these repositories to December 2023. Following a rigorous screening process, we identified and included 12 papers, encompassing a total of 994 subjects.

### Results

The meta-analysis unveiled a compelling overall effect size of exercise interventions on depressive symptoms in middle-aged and older individuals, represented by SMD = -0.41, with a 95% confidence interval of [-0.60, -0.22], and P < 0.0001. This statistical evidence underscores the significant and positive impact of exercise interventions on ameliorating depressive symptoms in this demographic. However, a degree of heterogeneity among studies was observed, with $I^2$ = 48% and P = 0.03.

### Conclusion

The comprehensive synthesis of outcomes elucidates the pronounced positive effect of exercise interventions in enhancing depressive symptoms among middle-aged and older adults. Specifically, the gentle and balanced exercise series emerges as particularly efficacious in mitigating depression. Furthermore, individual exercises stand out as more effective, with optimal results noted for moderate exercise sessions lasting 30 to 60 minutes. Our findings also highlight the superiority of short-term interventions, followed by medium- and long-term exercise interventions, in terms of efficacy. Nevertheless, recognizing the inherent

that all references are publicly available, thus ensuring the transparency and reproducibility of the study.

**Funding:** The author(s) received no specific funding for this work.

**Competing interests:** The authors have declared that no competing interests exist.

heterogeneity and potential limitations of our study, we advocate for future large-scale, comprehensive investigations to validate these findings. Additionally, optimizing exercise intervention protocols necessitates a more nuanced understanding, urging further research endeavors to refine strategies aimed at improving depressive symptoms in middle-aged and older adults.

# 1 Background

As demographic shifts accelerate the ageing of populations worldwide, the mental health of those in middle age and beyond has emerged as a critical area of public health concern. Depression, in particular, commands attention due to its considerable prevalence among older adults and its profound implications for both individuals and society at large. According to estimates from the World Health Organization (WHO), approximately 7% of the global middle-aged and elderly population is affected by depression. This prevalence exhibits significant variation across different countries and regions, mirroring the vast diversity in economic conditions, cultural norms, and social structures [1]. The impact of depression extends far beyond mood disturbances and a diminished quality of life. It is intricately linked with an array of physical health challenges. Among older adults, depression has been consistently associated with cognitive decline, an elevated risk of cardiovascular disease, and the aggravation of various chronic conditions. Furthermore, it can lead to a marked decline in social functionality, evidenced by increased isolation and loneliness, thereby intensifying the progression and repercussions of the condition [2].

Noteworthy is the higher incidence of depression among middle-aged and older women, likely attributable to unique stressors tied to changes in physical health, psychological wellbeing, and social roles. However, it is the elevated suicide rates among men within this demographic that underscore the pressing need for the timely identification and intervention of depression in this population, highlighting a gender disparity that demands targeted research and healthcare strategies [3]. This issue has been further compounded by the COVID-19 pandemic, which, through enforced isolation and social restrictions, has notably escalated depression risks among the elderly, further straining their already delicate mental health. While traditional medication remains a cornerstone in treating depression, its utility in middle-aged and older populations is marred by concerning side effects [4]. Given that these age groups frequently grapple with multiple chronic conditions, the interactions of medications and their effects on existing health issues are particularly worrisome [5]. For instance, antidepressants might aggravate cardiovascular issues, increase fall risk [6], or induce cognitive deterioration. Moreover, the slowed drug metabolism in older age leads to drug accumulation in the body, elevating side effect risks [7, 8]. These considerations underscore the need for caution in applying pharmacotherapy in middle-aged and elderly demographics and have spurred interest in alternative therapeutic approaches. In this context, exercise emerges as a promising non-pharmacological intervention. Garnering substantial interest from both scientific and clinical communities, exercise offers a low-risk avenue with extensive health benefits, presenting a valuable alternative in managing depression among the middle-aged and elderly [9].

In the realm of depression management, traditional pharmacotherapy, despite its central role, raises significant concerns regarding side effects, particularly within the middle-aged and elderly populations. These age groups often present with a constellation of chronic conditions, rendering the interplay between medications and their impact on pre-existing health issues a

critical area of focus [10]. Antidepressants, for instance, could potentially worsen cardiovascular conditions, amplify the risk of falls, or precipitate cognitive decline. Moreover, the age-related deceleration in drug metabolism may lead to an accumulation of pharmaceuticals in the body, thereby heightening the risk of adverse effects. These factors critically underscore the importance of evaluating the applicability and safety of pharmacological treatments in these demographics and have catalyzed the exploration of alternative therapeutic strategies. Exercise, as a non-pharmacological intervention, has garnered considerable acclaim from both the scientific community and clinical practitioners, owed to its minimal risk profile and multifaceted health benefits [11]. The therapeutic potential of exercise in ameliorating depression is multi-factorial. It modulates neurotransmitters, such as serotonin and endorphins, pivotal in mood regulation and mental health enhancement [12, 13]. Furthermore, regular physical activity boosts cerebral blood circulation, augmenting oxygen and nutrient delivery to the brain, thereby promoting neuronal health and neurogenesis. Exercise also plays a role in attenuating inflammation levels, a factor increasingly recognized in the pathophysiology of depression [14, 15]. Psychologically, it fosters a sense of self-efficacy and enhances social interactions among middle-aged and older adults.

Empirical evidence suggests the efficacy of various exercise forms and intensities in mitigating depressive symptoms in these age groups [16]. Moderate aerobic exercises like brisk walking, swimming, and cycling not only bolster cardiorespiratory fitness but also positively influence mood regulation [17, 18]. Similarly, strength training has demonstrated benefits for both mood and cognitive function [19, 20]. Crucially, it's imperative that these activities are tailored to be enjoyable and comfortable, ensuring adherence and pleasure in the process [21]. Practices such as gentle yoga and tai chi extend beyond physical wellness, offering mental relaxation and serving as potent tools in combatting depression [22, 23].

In the intricate landscape of research on exercise and depression, several studies have contributed nuanced insights. Jesper Krogh, Siri Kvam, and colleagues observed that physical activity can effectuate a moderate to large reduction in depressive symptoms, albeit the impact diminishes and loses significance over time. Notably, exercise's efficacy appeared diminished when compared with psychotherapy or antidepressant medication, suggesting its limited standalone antidepressant potency. Yet, as an adjunct to medication, exercise demonstrated moderately significant effects [24, 25]. Xiang Wang and collaborators found that physical activity notably lessened depressive symptoms in adolescents with moderate effectiveness, The study underscored the variability of exercise forms in achieving these outcomes [26]. Ye Hoon Lee, Yu Meng Xie, et al., in their research, highlighted the moderate efficacy of physical activity in alleviating depression, particularly in the adult demographic aged 18–64. They noted that higher exercise levels correlated with greater functional improvements [27, 28]. Lara Carneiro and team's investigation into resistance training revealed some positive impacts on depressive symptoms, albeit with substantial study-to-study heterogeneity [29]. This finding underscores the necessity for more rigorously designed randomized controlled trials to thoroughly evaluate resistance training as a potential depression treatment. Felipe B. Schuch and colleagues synthesized these findings, concluding that exercise significantly alleviates depressive symptoms. They emphasized the effectiveness of combined aerobic and anaerobic exercises, moderate-intensity activities, group exercises, and supervised mixed exercise forms, especially in treating depression among older adults [30]. Lijun Wang and team's study underscored the positive influence of physical activity on the psychological well-being of older adults, particularly effective in those with hypertension. However, they noted significant variances based on exercise type, duration, frequency, timing, and measurement tools. The study calls for more high-quality research to consolidate these findings [31]. Finally, Zichao Chen et al. concluded that low-intensity aerobic exercise and mind-body exercises are especially efficacious in mitigating

depressive symptoms in older adults, providing a compelling case for these exercise modalities in this population [32].

The existing body of literature indicates that moderate exercise might exert a beneficial impact on depressive symptoms. Nonetheless, this field of research is beset by methodological constraints, including limited sample sizes, suboptimal study designs, and inconsistencies in evaluating interventions and their outcomes. These limitations have cast a shadow of uncertainty over the efficacy of exercise interventions, particularly in the context of middle-aged and older populations. Moreover, the distinctive physiological, psychological, and social attributes of older adults may render their response to exercise interventions distinct from other age groups, necessitating more comprehensive research. The objective of our study is to conduct a systematic review and meta-analysis to investigate the effects of exercise interventions on depression in middle-aged and older adults. We aim to meticulously dissect the influence of various exercise types and intensities on treatment outcomes. Through this synthesis of existing evidence, we aspire to illuminate the efficacy of exercise interventions in alleviating depressive symptoms within this demographic, while delving into the underlying physiological and psychological mechanisms. This research endeavor seeks to broaden the spectrum of non-pharmacological options available for treating clinical depression. Concurrently, it aims to underpin public health strategies aimed at enhancing the mental health of middle-aged and older individuals by delineating the benefits and practical aspects of exercise interventions. The potential significance of our study extends beyond merely augmenting the quality of life for older adults; it also aims to offer empirical backing for health policymakers in formulating more efficacious health interventions, particularly in addressing the challenges posed by the global phenomenon of an aging population.

## 2 Methods

This meta-analysis was performed according to the Preferred Reporting Items for Systematic Reviews and Meta-Analysis statement and the Cochrane Collaboration Handbook. The protocol was registered on PROSPERO (CRD42023461736).

### 2.1 Data sources and searches

The systematic search was conducted by two independent reviewers (ZZ and LJY) in four databases: the Cochrane Library, Embase, PubMed and Web of Science, and was designed to retrieve articles up to Dec 2023, with disagreements resolved by consensus and by a third reviewer (ZKT) in case of disagreement. Terms from the Medical Subject Headings (MeSH) and words from the text were used as follows: ("Elderly" OR "Elderly" OR "Old" OR "Senior" OR "prenatal" OR "Mature" OR "Older" OR "Elderly person" OR "Aged person" OR "Senior citizen" OR "Elderly individual" OR "Advanced in years" OR "Geriatric" OR "Over the hill" OR "Elderly people" OR "Older generation" OR "Senior person") AND ("Depression" OR "Depressive Symptoms" OR "Depressive Symptom" OR "Symptom, Depressive" OR "Emotional Depression" OR "Depression, Emotional") AND ("Exercise" OR "Exercises" OR "Physical Activity" OR "Activities, Physical" OR "Activity, Physical" OR "Physical Activities" OR "Exercise, Physical" OR "Exercises, Physical" OR "Physical Exercise" OR "Physical Exercises" OR "Acute Exercise" OR "Acute Exercises" OR "Exercise, Acute" OR "Exercises, Acute" OR "Exercise, Isometric" OR "Exercises, Isometric" OR "Isometric Exercises" OR "Isometric Exercise" OR "Exercise, Aerobic" OR "Aerobic Exercise" OR "Aerobic Exercises" OR "Exercises, Aerobic" OR "Exercise Training" OR "Exercise Trainings" OR "Training, Exercise" OR "Trainings, Exercise") Specific details of the search algorithms for each database are provided in S1 File.

## 2.2 Inclusion and exclusion

In accordance with the PICOS framework integral to Cochrane's systematic evaluation [33], the selection criteria for literature inclusion are meticulously defined: **Participants (P):**The focus is on middle-aged and elderly individuals diagnosed with depression, aged 45 and above. The diagnosis must align with established diagnostic standards. **Intervention (I):**A spotlight on diverse multifactorial exercise interventions, characterized by varying content, intensity, duration, frequency, and cycles. **Comparison (C):**These interventions are benchmarked against standard care or placebo treatments. **Outcomes (O):**The principal outcome is the amelioration of depressive symptoms, quantifiable through standardized tools like the Hamilton Depression Scale, Hamilton Depression Rating Scale (HDRS), or the Beck Depression Inventory (BDI). Secondary outcomes extend to life quality, sleep quality, and cognitive function. **Study Design (S)**: The primary reliance is on Randomized Controlled Trials (RCTs), renowned as the pinnacle of clinical effectiveness research.

Contrastingly, the exclusion criteria are: 1. Studies not focused on the specified demographic of middle-aged and elderly patients with depression. 2. Research centered on medication, psychotherapy, or other non-exercise-based interventions. 3. Non-original studies, including commentaries, case reports, expert opinions, and reviews. 4. Studies with prominent methodological shortcomings, such as insufficient sample sizes, absence of control groups, or unclear randomization procedures. 5. Literature not published in English. 6. Studies not conducted as RCTs.

## 2.3 Assessment of risks of bias

In adherence to the esteemed Cochrane Collaboration guidelines, reviewers ZZ and LJY independently conducted a meticulous risk of bias assessment for each study included in our analysis. The Cochrane Collaboration's guidelines represent a comprehensive and universally recognized framework, enabling a standardized evaluation of the methodological integrity of each study. This detailed assessment process, ingrained in the Cochrane Collaboration ethos, involves a granular examination of potential biases. Key areas scrutinized include randomization processes, allocation concealment, the blinding of participants and researchers (notably challenging in certain ethical contexts), completeness of outcome data, risks of selective reporting, and other potential biases. Each study's evaluation was independently undertaken by the reviewers, with discrepancies resolved through discussion or, when necessary, by consulting a third reviewer (ZKT).The outcomes of this rigorous bias risk assessment are meticulously compiled in a risk of bias table, which is available in S2 File. This table provides a transparent and comprehensive overview of the methodological quality of each study considered. Importantly, when interpreting the results of the meta-analyses and forming conclusions regarding the efficacy of exercise interventions for depression in middle-aged and older adults, these bias risk assessments were carefully factored into our deliberations. This approach ensures a balanced and methodologically sound interpretation of the data, upholding the high standards of evidence synthesis championed by the Cochrane Collaboration.

## 2.4 Data extraction

With a standardized form, two reviewers (ZZ and LJY) independently extracted the pertinent data from each included study, encompassing essential details such as author names, year of publication, sample size of the intervention and control groups, age group characteristics of both intervention and control groups, type of intervention, intervention length, frequency, and duration, type of control group, and outcome measures. The utilization of a standardized form ensured consistency and accuracy in data extraction across all studies, minimizing the

risk of errors and enhancing the reliability of the collected information. Each reviewer diligently recorded the required data elements from the eligible studies, and any discrepancies or uncertainties were resolved through discussion or consultation with a third reviewer (ZKT) if necessary. By employing this rigorous and systematic data extraction approach, we obtained comprehensive and reliable information from the included studies, forming the foundation for our comprehensive META analysis. The detailed data extracted from each study are presented in the supplementary materials, providing transparency and facilitating a thorough understanding of the primary characteristics of the studies included in our research.

## 2.5 Assessment of overall effect size

Statistical analyses were conducted using Review Manager V.5.3, and overall effect sizes were calculated based on the statistical analyses of the results from the measurement scale tests of the twelve included articles. Hedge's g standardised effect sizes were utilized for each included study to measure the intervention's effect size, with effect sizes of 0.2, 0.4, and 0.8 indicating small, medium, and large effects, respectively. To ensure consistency and that all effect sizes were in the expected direction of the intervention, $p < 0.05$ was considered significant. Given that there are different measures of the effect of exercise on depression in middle-aged and older adults, the standardised mean difference (SMD) was chosen as it reflects the overall intervention effect size. To synthesize the effect of physical activity on depression scores in middle-aged and older adults in the meta-analysis, the standardised mean difference (SMD) was calculated using the Practical Meta-Analysis Effect Size Calculator (Wilson) along with its corresponding 95% confidence intervals. A heterogeneity test was also conducted to assess the extent of differences between the included studies in describing the overall effect sizes. Heterogeneity was assessed using methods such as the Q statistic and the $I^2$ indicator. $I^2$ values quantitatively assessed heterogeneity, where 0% indicated no heterogeneity, $\geq 25\%$ indicated low heterogeneity, $\geq 50\%$ indicated moderate heterogeneity, and $\geq 75\%$ indicated high heterogeneity. When $I^2$ values indicated moderate to high heterogeneity, a random-effects model was used for data combination, and conversely, a fixed-effects model was utilized.

## 2.6 Subgroup analysis of exercise intervention programmes

To elucidate the sources of heterogeneity and to unravel the nuanced effects of diverse exercise intervention programmes on middle-aged and older patients with depression, this study embarks on a series of subgroup analyses. The intent is to discern which exercise modalities yield the most significant therapeutic benefits, thereby furnishing clinicians with refined, evidence-based recommendations. The subgroup analyses are meticulously categorized into the following dimensions:

1. Individual vs. Group Exercise: This analysis focuses on contrasting the impact of solo exercise interventions against those conducted in group settings, specifically targeting the improvement in depressive symptoms among the middle-aged and elderly demographic.

2. Classification of Exercise Forms: The exercise modalities are systematically classified into four principal categories: balance series, vitality aerobic series, strength building series, and interactive technology fitness series.

3. Individual Exercise Duration: This pertains to evaluating how varying durations of individual exercise sessions influence the effectiveness of depression interventions.

4. Total Exercise Duration: This segment analyses the effect of the cumulative duration of the entire intervention cycle (spanning weeks to months) on the therapeutic efficacy.

For these subgroup analyses, pertinent data were extracted from the included studies. Using Review Manager V.5.3, we computed effect sizes (Hedge's g) along with their 95% confidence intervals for each subgroup, assessing significance at p<0.05. Heterogeneity tests were employed to ascertain significant differences between subgroups. Central to our analyses is the impact of exercise interventions on mitigating depressive symptoms [34, 35]. The insights garnered from these subgroup analyses will deepen our understanding of the specific effects of varied exercise protocols on our targeted patient group. These findings will not only offer more targeted clinical recommendations but also fuel a detailed discussion in the subsequent sections of our study. This will, in turn, chart a course for future research and practical applications in this domain, reinforcing the intersection of exercise science and mental health treatment for the aging population.

## 3 Results

### 3.1 Search process

In our systematic search across four databases, we initially identified 3,417 studies. The process of deduplication pared this number down to 2,694 unique studies. These were then subjected to a rigorous screening based on titles and abstracts, which further refined our pool to 238 studies warranting full-text evaluation. During the detailed analysis of these 238 studies, a majority, amounting to 226, were excluded. The primary reasons for exclusion were their lack of relevance to the target demographic of middle-aged and older individuals with depression, non-compatibility of interventions, and mismatch in outcome indicators.

Consequently, our meta-analysis was distilled to a core collection of 12 trials, each contributing valuable data. The meticulous methodology we employed from the initial database query to the final selection of these trials is systematically depicted in Fig 1.

### 3.2 Characteristics of the included studies and participants

Table 1 delineates the characteristics of the 12 trials incorporated into our meta-analysis. These trials span publications from 2001 to 2023. The sample sizes of the included studies varied, ranging from 21 to 247 participants. Table 1 comprehensively details the sample sizes for both intervention and control groups in each study, alongside demographic information such as mean age and male-to-female ratios. Furthermore, the table categorizes the types of exercise interventions employed in these studies, which include yoga, tai chi, walking, dancing, resistance training, and interactive fitness programs based on video games. The duration of these exercise interventions varied considerably, extending from 3 weeks to 6 months, with the frequency of these interventions ranging from once to four times per week.

Our analysis also focused on a diverse array of depression outcome measures. These measures encompassed tools such as the Geriatric Depression Scale (GDS), its short form in the Korean version (GDSSF-K), the Hospital Anxiety and Depression Scale (HADS), the Self-Rating Depression Scale (SDS), the Taiwanese Depression Questionnaire (TDQ), and the Beck Depression Inventory (BDI). The selection of these scales ensures a comprehensive assessment of depressive symptoms across various cultural and demographic contexts.

### 3.3 Risks of bias

In our meta-analysis, the risk of bias across the 12 included papers was systematically assessed across several dimensions. All 12 papers demonstrated a low risk in terms of random sequence generation. In the context of allocation concealment, 7 papers were classified as low risk, while 5 papers presented an unknown risk due to insufficient reporting in this area. Regarding participant blinding, a key factor in clinical trials, our assessment revealed that 9 papers were

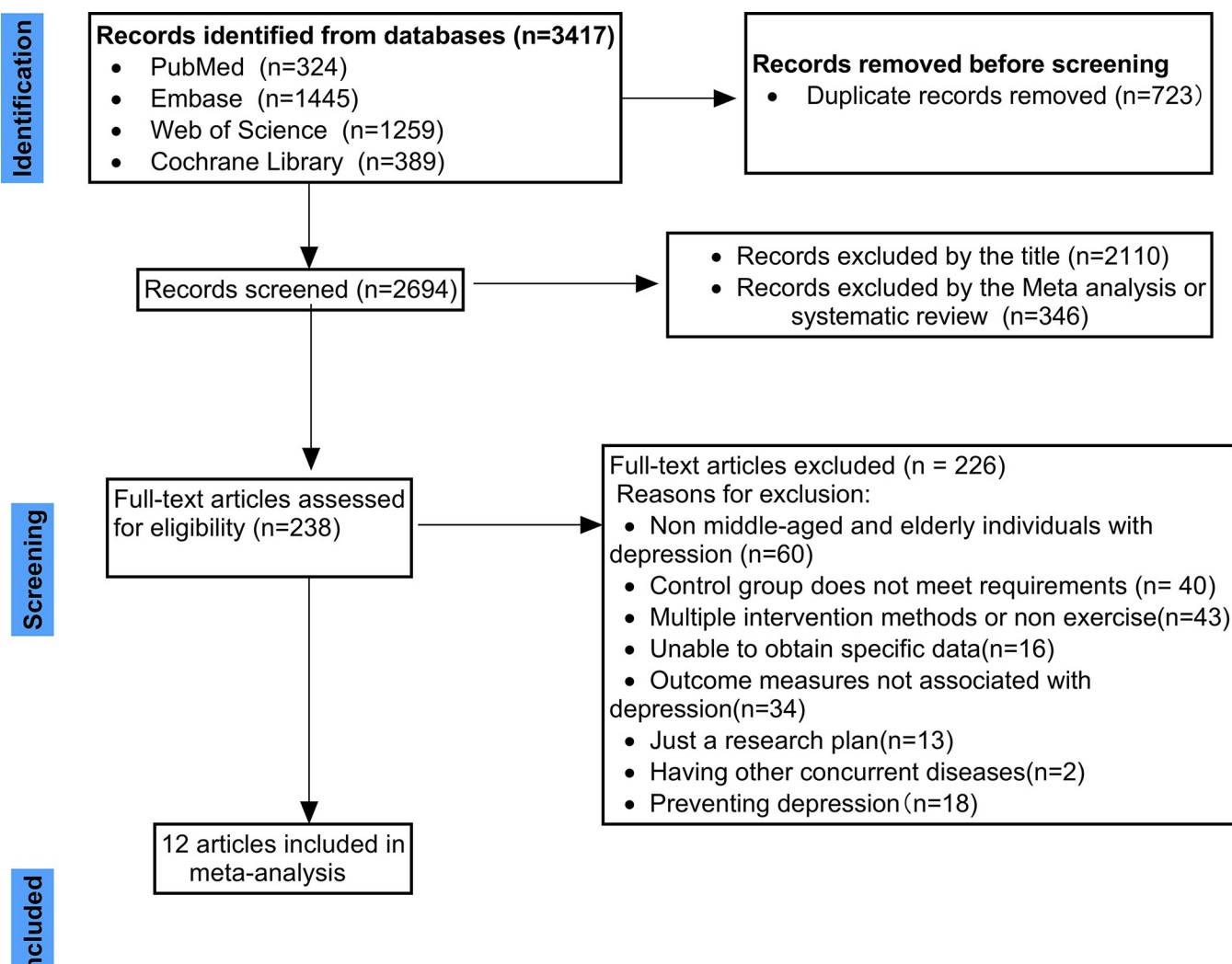

**Fig 1. Flowchart and selection of studies, ensuring transparency and replicability of our research process.**

categorized as high risk, 2 as unknown risk (attributable to underreporting), and only 1 paper was considered low risk. Similarly, for evaluator blinding, 5 instances were deemed low risk, 5 were unclear due to underreporting, and 2 were high risk. The risk assessment for assessor blinding showed a similar trend, with 5 studies rated as low risk, 5 as unknown (due to under-reporting), and 2 as high risk. For the dimension of 'incomplete outcome data', 10 studies were evaluated as low risk, and 2 were marked as unknown risk due to underreporting. The evaluation of selective reporting bias yielded 6 studies each in the categories of low risk and unknown risk, the latter again due to inadequate reporting. In terms of other potential biases, 11 studies were assessed as unclear risk, primarily due to the absence of necessary information. A detailed assessment of these risks of bias is comprehensively presented in S2 File, providing a transparent and rigorous overview of our evaluative methodology.

## 3.4 Meta-analysis

**3.4.1 Baseline period test.** Prior to the execution of our meta-analysis, we conducted a preliminary test to assess baseline variability in the selected studies. This was a critical step to

**Table 1. Characteristics of the included studies and participants.**

| Studies | Sample Size (IG/CG) | Age Range (IG/CG) | Gender ratio (IG/CG)M:F | IG Type | CG Type | Frequency/duration | outcome measures |
|---|---|---|---|---|---|---|---|
| Min-Jung Choi 2017 | 33/30 | 77.6 ± 5.69/78.8 ± 5.83 | 3:30 /1: 29 | sitting yoga | usual care | 4 times weekly/12 weeks | GDSSF-K |
| Mei-Lan Chen 2023 | 14/14 | 76.79 ± 4.79/78.57 ± 5.44 | 3:11/3:11 | Resistance Exercise | usual activities | 2 times weekly/12 weeks | GDS |
| Yun-Sik Kim 2018 | 11/10 | 76.10 ± 3.85/76.40 ± 3.27 | 0:11/0:10 | strength training | usual activities | 3 times weekly/24 weeks | GDSSF-K |
| Kyeongjin Lee 2023 | 28/29 | 80.39 ± 2.57/79.10 ± 3.90 | 17:11/14:15 | Home-Based Exergame | usual activities | 3 times weekly/8 weeks | GDS |
| S. Aguiˉnaga 2018 | 124/123 | 70.62±5.02/71.43 ±5.25 | 45:113/26:123 | Home-Based Physical Activity | attentional control | /6 months | HADS |
| S. J. Liao 2018 | 55/52 | 71.84 ± 7.297/71.75 ± 8.201 | 19:36/22:30 | Tai Chi | usual activities | 3 times weekly/3 months | GDS |
| Yuko Kai 2016 | 20/20 | 51.0±7.0/51.2±7.9 | 0:20/0:20 | Stretching | usual activities | once a day/3 weeks | SDS |
| Liang Hu 2017 | 40/40 | 52.60±4.12/54.15 ±2.32 | 0:40/0:40 | Walking | usual activities | 3 times weekly/16 weeks | BDI |
| Jane Sims 2006 | 14/18 | 75.25±5.78/74.30 ±5.72 | 2:12/9:9 | strength training | usual activities | 3 times weekly/10 weeks | GDS |
| Kuei-Min Chen 2009 | 62/66 | 69.20± 6.23 | | silver yoga | usual activities | 3 times weekly/6 months | TDQ |
| NalinA. Singh 2001 | 15/14 | 71±2 | | strength training | usual activities | 3 times weekly/10 weeks | BDI |
| H Vankova 2014 | 79/83 | 83.38±8.23/82.85 ±7.87 | 3:76/10:73 | Dance | usual activities | 1 time weekly/3 months | GDS |

Note

Abbreviations: IG = Intervention group; CG = Control group; BDI = Beck depression inventory; HADS = Hospital Anxiety Depression Scale; GDS = Geriatric Depression Scale; GDSSF-K = short form in the Korean version; SDS = Self-Rating Depression Scale; TDQ = Taiwanese Depression Questionnaire

ascertain whether substantial differences existed between the experimental and control groups at the outset of the exercise interventions. Such baseline assessment is imperative to ensure that the evaluation of effect sizes is not confounded by initial discrepancies. Given the use of various instruments for depression assessment in the literature, including BDI-II, CES-D, HADS, and SDS, we opted to analyze the standardized mean difference (SMD) to harmonize these diverse scales. This approach was aimed at providing a more accurate and comparable measure of depression levels across different studies. Our findings indicated an SMD of 0.17261 between the experimental and control groups at baseline, with 95% confidence intervals ranging from -0.001357 to 0.346583. Although the confidence intervals bordered on non-zero, the combined effect sizes showed a statistical significance boundary of $p = 0.052$. Furthermore, the heterogeneity test revealed no significant discrepancies ($\chi^2 = 18.06$, df = 11, $p = 0.080$), suggesting consistent baseline results across the included studies, as depicted in Fig 2. This uniformity in baseline depressive symptoms among participants across the 12 studies provides a robust foundation for our subsequent analysis. It allows us to integrate the results from individual studies more confidently, estimating the overall impact of exercise interventions on middle-aged and older individuals with depression. With minimal heterogeneity observed between the studies, our meta-analysis positions us to explore the viability of exercise interventions as a potentially effective component in treatment programs for depression in this demographic. Consequently, the collective results of these trials bolster our investigation into the potential therapeutic benefits of exercise for depression in older adults.

```
              Study |      SMD   [95% Conf. Interval]     % Weight
----------------+---------------------------------------------------
M J Choi        |   .416364    -.083551    .916279      8.00377
M L Chen        |   .746345    -.021724    1.51441       4.21434
H Vankova       |   .242622    -.066595    .551839       13.521
Y S Kim         |  -.084851    -.941645    .771944       3.50977
K J Lee         |  -.028026     -.54734    .491287       7.60608
S Agui~naga     |  -.011991    -.261414    .237433       15.8926
S J Liao        |   .021286    -.357827      .4004       11.1289
Y Kai           |  -.336555    -.960951    .287842       5.84168
L Hu            |   .125025    -.313675    .563725       9.43781
J Sims          |   .118187    -.580882    .817257       4.90588
K M Chen        |   .236631    -.111244    .584505       12.143
N A Singh       |   1.39665      .57857    2.21473       3.79517
----------------+---------------------------------------------------
  D+L pooled SMD |   .172613    -.001357    .346583
----------------+---------------------------------------------------
  Heterogeneity chi-squared =  18.06 (d.f. = 11) p = 0.080
  Estimate of between-study variance Tau-squared =  0.0334
  Test of SMD=0 : z= 1.94 p = 0.052
```

**Fig 2. Plot of results of baseline difference test.**

**3.4.2 Meta-analysis result.** In our meta-analysis, we scrutinized the impact of exercise interventions on depressive symptoms in middle-aged and older individuals. This involved aggregating data from 495 participants in the experimental group and 499 in the control group. To accommodate potential study-to-study variability, we employed a random-effects model for calculating effect sizes. The analysis yielded a pooled standardized mean difference (Std. Mean Difference) of -0.41, with a 95% confidence interval spanning from -0.60 to -0.22, as illustrated in Fig 3. The negative value here signifies a positive outcome, indicating a notable reduction in depressive symptoms in the exercise intervention group compared to the control.

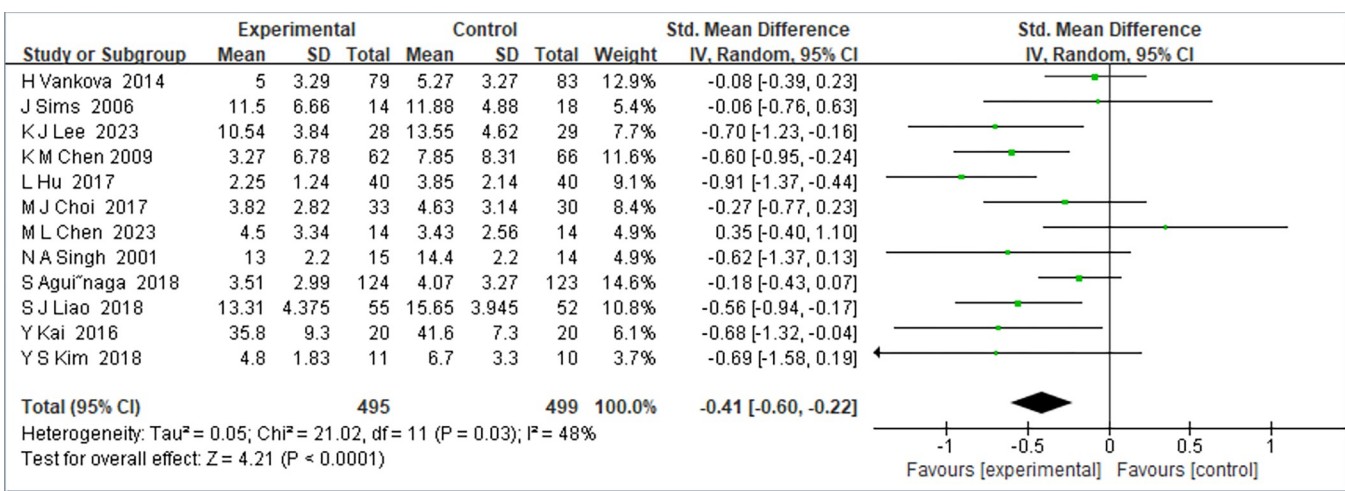

**Fig 3. Forest plot of the effect of exercise intervention on depression in middle-aged and older adults.**

The absence of zero in the confidence interval further solidifies the significance of this effect. However, the heterogeneity analysis revealed a moderate $I^2$ value of 48%, suggesting notable variability among study results. This variability could stem from differences in experimental designs, participant characteristics, depression assessment tools, or the nature and intensity of the interventions.

To investigate the origins of this heterogeneity, we conducted sensitivity analyses, removing individual studies to gauge their impact on overall effect size. The persistence of heterogeneity, despite the removal of specific studies, implies that it might be attributed to factors beyond the influence of any single study. Further scrutiny indicated that the heterogeneity might be linked to the diverse instruments used to measure depressive symptoms, as well as the focus of some trials on menopausal women. The studies in our meta-analysis employed five different instruments, including BDI-II, CES-D, HADS, SDS, and TDQ, for depression assessment. The varying sensitivity and specificity across these tools could have led to differential effects. Furthermore, the inclusion of trials specifically targeting menopausal women—a group potentially exhibiting unique physiological, psychological, and social characteristics—might also have influenced the consistency of the results [36–38]. These factors are crucial for interpreting the findings of our meta-analysis, as they highlight the complexity and nuanced nature of evaluating exercise interventions' efficacy in alleviating depressive symptoms among older populations.

Despite the identified sources of heterogeneity, current evidence robustly supports the efficacy of exercise interventions in treating depression among middle-aged and older patients. This finding is significant, as it underscores the therapeutic potential of physical activity in managing depressive symptoms in this demographic. To enhance the precision and applicability of exercise intervention programs, future research endeavors should focus on a couple of key areas. First, there is a need for standardization in the measurement of depressive symptoms. By harmonizing these measures, studies can yield more directly comparable and consistent results, thereby improving the reliability and validity of the findings. Second, it is imperative to delve into the specific requirements of diverse population subgroups, such as menopausal women. This approach recognizes that different groups may have unique physiological, psychological, and lifestyle factors influencing their response to exercise interventions. Understanding and addressing these specific needs will not only refine intervention strategies but also ensure that treatment outcomes are more tailored and effective. In conclusion, while the positive role of exercise interventions in treating depression in older adults is evident, refining and tailoring these interventions through focused research will further optimize their effectiveness and applicability across varied population segments.

**3.4.3 Publication bias test.**   In our meta-analysis, we rigorously evaluated the potential for publication bias, a critical step in ensuring the validity of our findings. Our initial approach involved a visual examination using Egger's publication bias plot, as depicted in Fig 4.

In this meta-analysis, we encountered a plethora of methodologies for evaluating potential publication bias. Initially, Egger's test emerged as a quantitatively robust statistical tool for identifying small study effects and scrutinizing publication bias. Its efficacy is particularly pronounced in the realm of continuous variable effect sizes. This stems from its capacity to investigate the correlation between effect sizes and their precision via regression analysis. Such an approach is instrumental in uncovering potential associations between the accuracy of studies and the effect sizes they report, especially since our meta-analyses predominantly centered on continuous data. Moreover, while Egger's test, alongside other publication bias assessment instruments such as Begg's test and the visual analysis of funnel plots, each possesses unique advantages, Egger's test is reputed for its heightened sensitivity in studies characterized by small sample sizes [39]. Acknowledging Egger's test's constraints, including its diminished

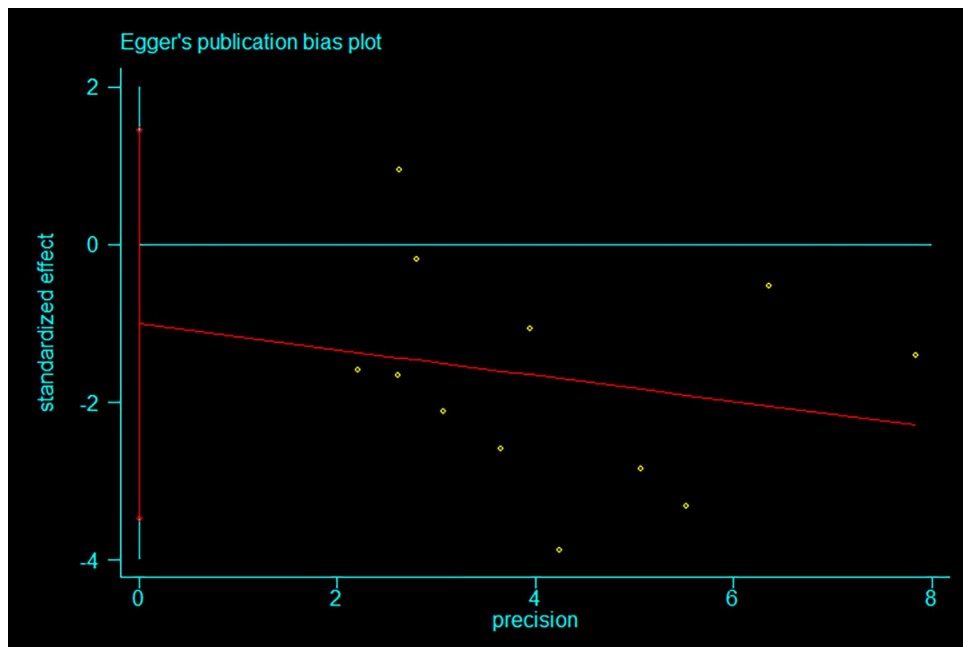

**Fig 4. Egger's publication bias plot.**

responsiveness to non-linear effect size alterations and the possibility of generating misleading outcomes in the presence of pronounced heterogeneity, we meticulously appraised our dataset and study methodology. This scrutiny led us to conclude that these limitations exert a minimal influence on our current investigation. Additionally, we explored alternative methods, such as the visual examination of funnel plots and Begg's test. Nevertheless, considering the distinct conditions and data nature of our study, Egger's test emerged as the most fitting option, offering an optimal equilibrium.

In this plot, we utilized the precision of the studies, defined as the inverse of the standard error, as the horizontal axis, while the standardized effect size was plotted on the vertical axis. Under ideal circumstances, where publication bias is absent, one would anticipate an even distribution of studies on both sides of the vertical line representing a zero effect size. This symmetrical distribution is indicative of a random spread of effect sizes, both positive and negative, across studies of varying precision. Upon inspection, our plot revealed a downward trajectory in standardized effect sizes with an increase in study precision. However, this observed trend, suggestive of potential bias, was not conclusively substantiated by subsequent statistical tests. While the visual pattern hints at a possible bias, the lack of statistical reinforcement implies that this observation should be interpreted with caution. This nuanced finding highlights the importance of combining visual assessments with statistical analyses when investigating publication bias. It is crucial to acknowledge that while visual plots can offer valuable insights, they are not definitive without the support of statistical evidence. Our careful approach in assessing publication bias reflects a commitment to thoroughness and rigor, essential in ensuring the reliability of meta-analytical results.

In our rigorous assessment for publication bias, we employed Egger's test, yielding detailed statistical results. The test produced a slope of -1.627795, with a 95% confidence interval ranging from -0.7115669 to 0.3860079. Crucially, this interval includes the zero point, indicating that the slope was not statistically significant, evidenced by a P-value greater than $|t|$ of 0.524. Furthermore, the detection bias estimate was calculated at -1.012169, with an associated t-

```
Egger's test
```

| Std_Eff | Coef. | Std. Err. | t | P>\|t\| | [95% Conf. Interval] | |
|---|---|---|---|---|---|---|
| slope | -.1627795 | .2462986 | -0.66 | 0.524 | -.7115669 | .3860079 |
| bias | -1.012169 | 1.104052 | -0.92 | 0.381 | -3.47215 | 1.447811 |

**Fig 5. Egger's test.**

value of -0.92. This resulted in a P-value of 0.381 and a 95% confidence interval from -3.47215 to 1.447811, which, again, encompasses the null value, as illustrated in Fig 5 These statistical findings provide robust evidence against the presence of publication bias in our dataset. Despite the initial visual trend observed in Egger's publication bias plot suggesting that smaller studies might report larger effect sizes, the statistical analysis using Egger's test did not confirm the significance of this trend. Consequently, we can assert with greater confidence that our analysis is not marred by significant publication bias. This enhances the reliability of our results, suggesting that the studies included in our meta-analysis represent a comprehensive sample. It implies that our findings are less likely to be influenced by systematic biases or the exclusion of unpublished research. This assurance of minimal publication bias reinforces the credibility of our meta-analysis, affirming its contribution to the understanding of exercise interventions in middle-aged and older patients with depression.

**3.4.4 Subgroup analysis.** In this investigation, we meticulously structured subgroups to dissect four pivotal aspects of the exercise regimen: the format (individual versus collective), the nature of the exercise intervention, the length of each exercise session, and the overall span of the intervention (refer to Table 2 for detailed categorization). The exercise interventions were categorized into distinct series: Balance Series, Vitality Aerobic Series, Strength Building Series, and the cutting-edge Interactive Science and Technology Fitness Series. Furthermore, the exercise session lengths were sub-divided into three categories: short-duration ($\leq$30 minutes), medium-duration (30 to 60 minutes), and long-duration (>60 minutes). Similarly, the duration of the exercise programmes was classified into short-term ($\leq$8 weeks), medium-term (>8 weeks to $\leq$16 weeks), and long-term (>16 weeks), providing a comprehensive framework for our systematic analysis.

**Table 2. Subgroup analysis of an exercise intervention for depressive symptoms in middle-aged and older adults.**

| Adjustment variables | Subgroup | Heterogeneity test | | Sample size | SMD [95%CI] |
|---|---|---|---|---|---|
| | | P | I$^2$(%) | | |
| | Collective | 0.65 | 0% | 600 | -0.36[-0.64,-0.08] |
| | Individual | | | 394 | -0.45[0.73, -0.17] |
| Type of exercise | Balance Series | 0.64 | 0% | 585 | -0.41[-0.62,-0.20] |
| | Vitality Aerobic Series | | | 242 | -0.48[-1.28,0.33] |
| | Strength Building Series | | | 110 | -0.23[-0.70,0.24] |
| | Interactive technology fitness | | | 57 | -0.70[-1.23,-0.16] |
| Exercise session lengths | short-duration | 0.75 | 0% | 40 | -0.68[-1.32,-0.04] |
| | medium-duration | | | 364 | -0.50[-0.80,-0.20] |
| | long-duration | | | 311 | -0.39[-0.81,0.03] |
| Duration of the exercise programmes | short-term | 0.38 | 0% | 97 | -0.69[-1.10, -0.28] |
| | medium-term | | | 501 | -0.34[-0.63, -0.04] |
| | long-term | | | 396 | -0.40[-0.75, -0.06] |

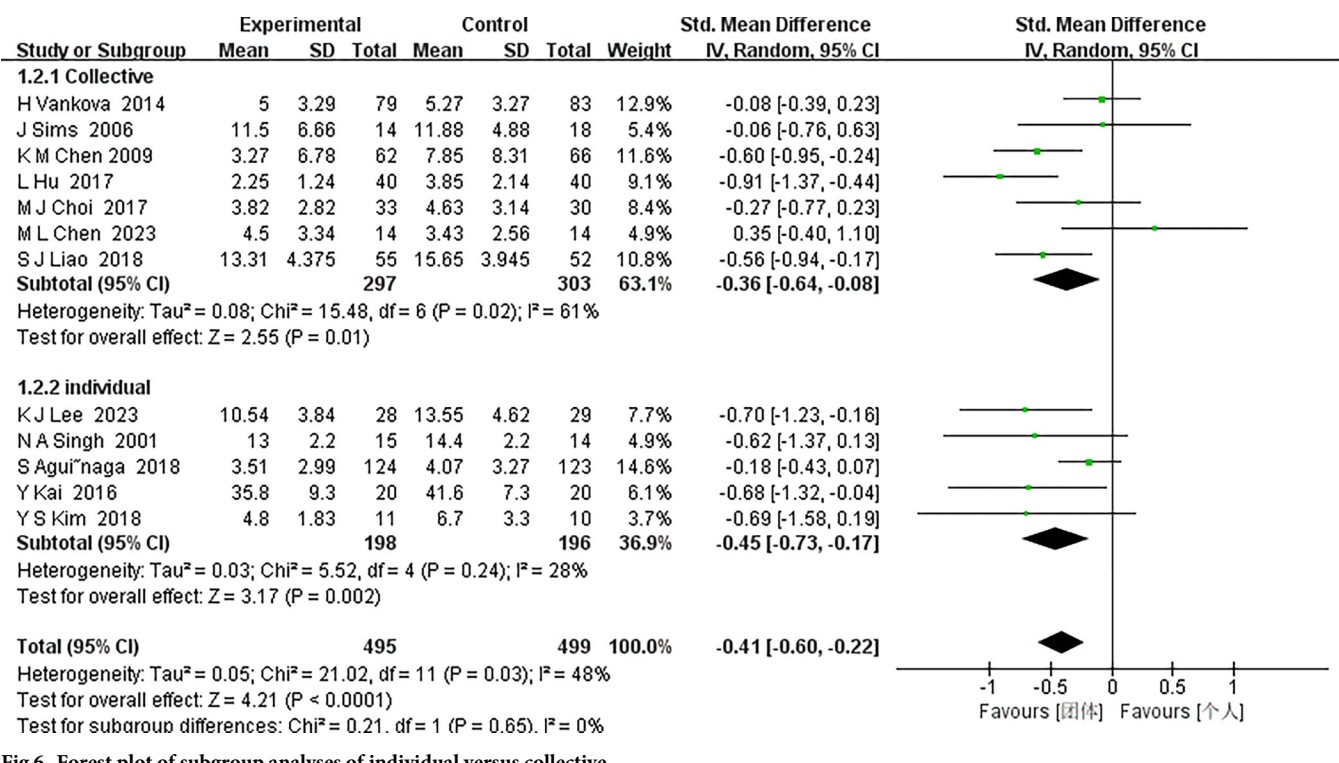

**Fig 6. Forest plot of subgroup analyses of individual versus collective.**

*3.4.4.1 Individual versus collective.* In this comprehensive meta-analysis, a random-effects model was meticulously adopted to accommodate the inherent heterogeneity observed across the incorporated studies. The studies were bifurcated into two distinct subgroups: group exercise interventions and individual exercise interventions, with an aim to elucidate any differential impacts of these interventions on middle-aged and older individuals grappling with depression. In the realm of group interventions, a moderate degree of heterogeneity was observed ($I^2$ = 61%). Despite this, the collective effect size (Standardized Mean Difference, SMD = -0.36, with a 95% Confidence Interval [CI] ranging from -0.64 to -0.08) manifested a significant antidepressant impact when juxtaposed with the control group. Conversely, the individual intervention subgroup exhibited a lower heterogeneity ($I^2$ = 28%). Here, the overall effect size (SMD = -0.45, 95% CI [-0.73, -0.17]) similarly underscored the efficacy of exercise in mitigating depressive symptoms, as delineated in Fig 6. A crucial observation was the lack of significant disparity between the two intervention types ($Chi^2$ = 0.21, degrees of freedom [df] = 1, P = 0.65), suggesting a comparable effectiveness of group and individual interventions in the context of exercise's impact on the target demographic. This parity infers that the beneficial influence of exercise interventions might be independent of the participation mode in middle-aged and elderly individuals with depression. When considering the overarching effect, the analysis revealed a notable overall efficacy of the exercise intervention (SMD = -0.41, 95% CI [-0.60, -0.22]), thus reinforcing the advantageous role of physical activity in this patient cohort. This aligns with previous research and bolsters the proposition of incorporating exercise as a therapeutic strategy for middle-aged and elderly individuals with depression. Nevertheless, it is imperative to acknowledge the discerned heterogeneity, particularly pronounced in the group intervention subgroup. This variability could emanate from disparities in participant demographics, the nature and specifics of the exercise intervention, and variations in exercise intensity and frequency. Future research endeavors should delve into how

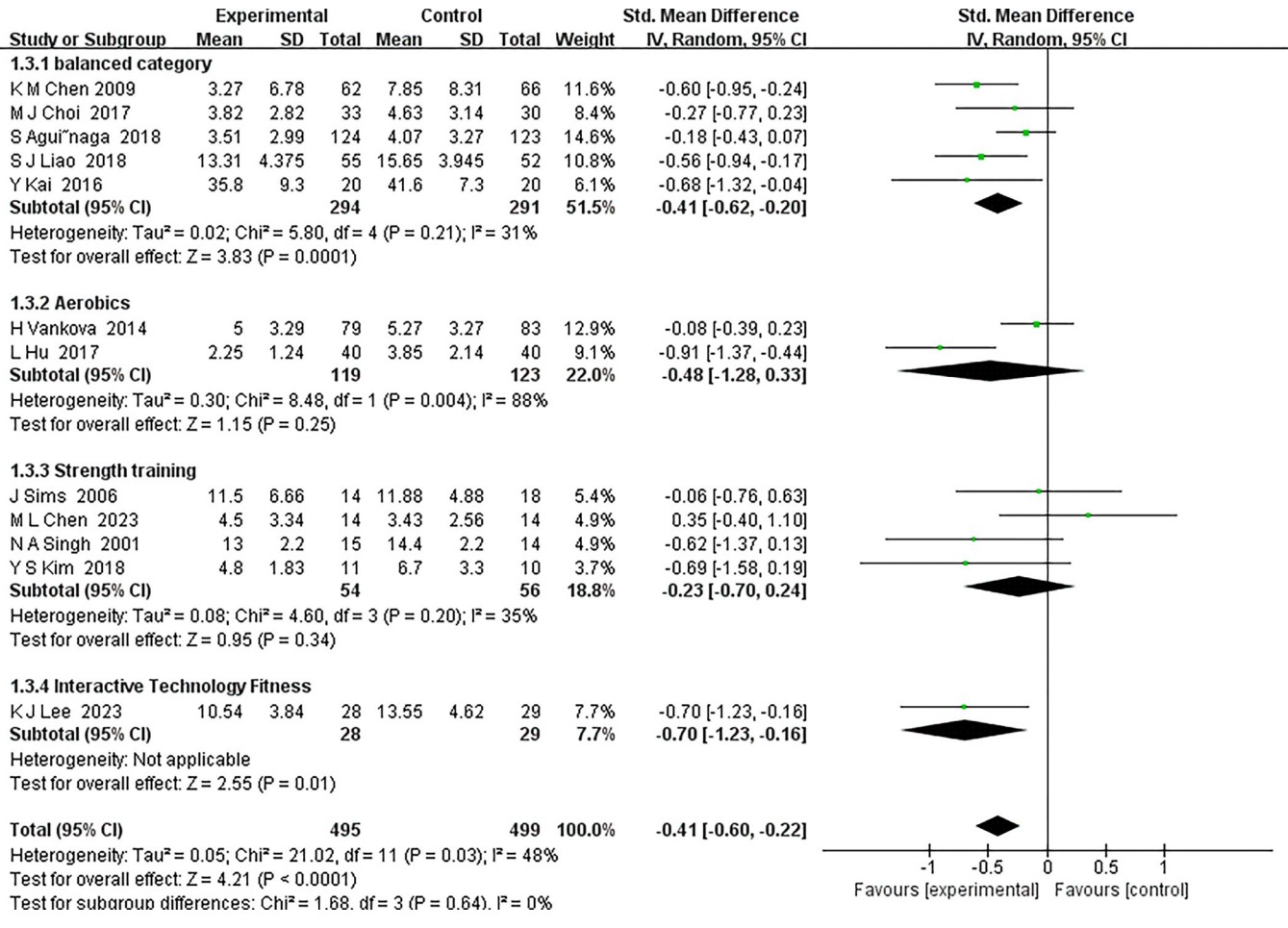

**Fig 7. Forest plot of subgroup analyses of type of exercise.**

these variables modulate the effectiveness of exercise interventions, with an ultimate goal of refining and optimizing these programmes for enhanced therapeutic outcomes.

*3.4.4.2 Type of exercise.* In this focused subgroup analysis, we meticulously evaluated the impact of exercise interventions on middle-aged and older patients with depression, categorizing them into four distinct subgroups based on the nature of the exercise regimen. These included the "Gentle Balance Series", "Vitality Aerobic Series", "Strength Building Series", and "Interactive Technology Fitness Series", each characterized by unique exercise modalities. The subsequent analysis, detailed in Fig 7, offers insightful findings:

1. Gentle Balance Series: This subgroup encompassed four studies focusing on exercises like yoga and Tai Chi. The interventions demonstrated a notable standardized mean difference (SMD) of -0.41 (95% CI: -0.62 to -0.20), signifying a significant positive influence on alleviating depressive symptoms. A noteworthy aspect was the low heterogeneity ($I^2$ = 0%), indicating a high consistency in the effects across studies.

2. Vitality Aerobic Series: Comprising two trials, this subgroup engaged participants in aerobic activities such as walking and dancing. The overall SMD was calculated at -0.48 (95% CI: -1.28 to 0.33). Although the midpoint hinted at a beneficial effect, the broad confidence interval suggests a degree of uncertainty. Remarkably, the heterogeneity in this subgroup was nonexistent ($I^2$ = 0%), pointing to a consistent effect across studies.

3. Strength Building Series: This subgroup included three studies involving exercises like resistance training. The overall SMD stood at -0.23 (95% CI: -0.70 to 0.24), tentatively indicating a positive effect, though the result remains inconclusive due to the wide confidence interval. This subgroup exhibited moderate heterogeneity ($I^2$ = 35%), suggesting some variability in the effects.

4. Interactive Technology Fitness Series: Based on a single study, this innovative subgroup utilized an interactive fitness program integrating video game elements. The SMD was -0.70 (95% CI: -1.23 to -0.16), proposing that such interventions may substantially alleviate symptoms of depression.

*3.4.4.3 Exercise session lengths.* In this study, a detailed subgroup analysis was conducted to elucidate the impact of exercise duration on ameliorating depressive symptoms in middle-aged and older patients. Grounded in prior research, the exercise durations were categorized into three subgroups: ≤30 minutes, 30 to 60 minutes, and >60 minutes. The findings, also illustrated in Fig 8, are summarized as follows:

1. Short-duration exercise subgroup (≤30 minutes):This subgroup, represented by a single study, indicated a significant positive effect of short-duration exercise on depressive symptoms, with a Standardized Mean Difference (SMD) of -0.68 and a 95% Confidence Interval (CI) of -1.32 to -0.04. This implies that even brief exercise sessions can yield beneficial therapeutic effects.

2. Moderate-duration exercise subgroup (30–60 minutes): Encompassing six trials, this subgroup demonstrated an SMD of -0.50 (95% CI: -0.80 to -0.20), suggesting that moderate-

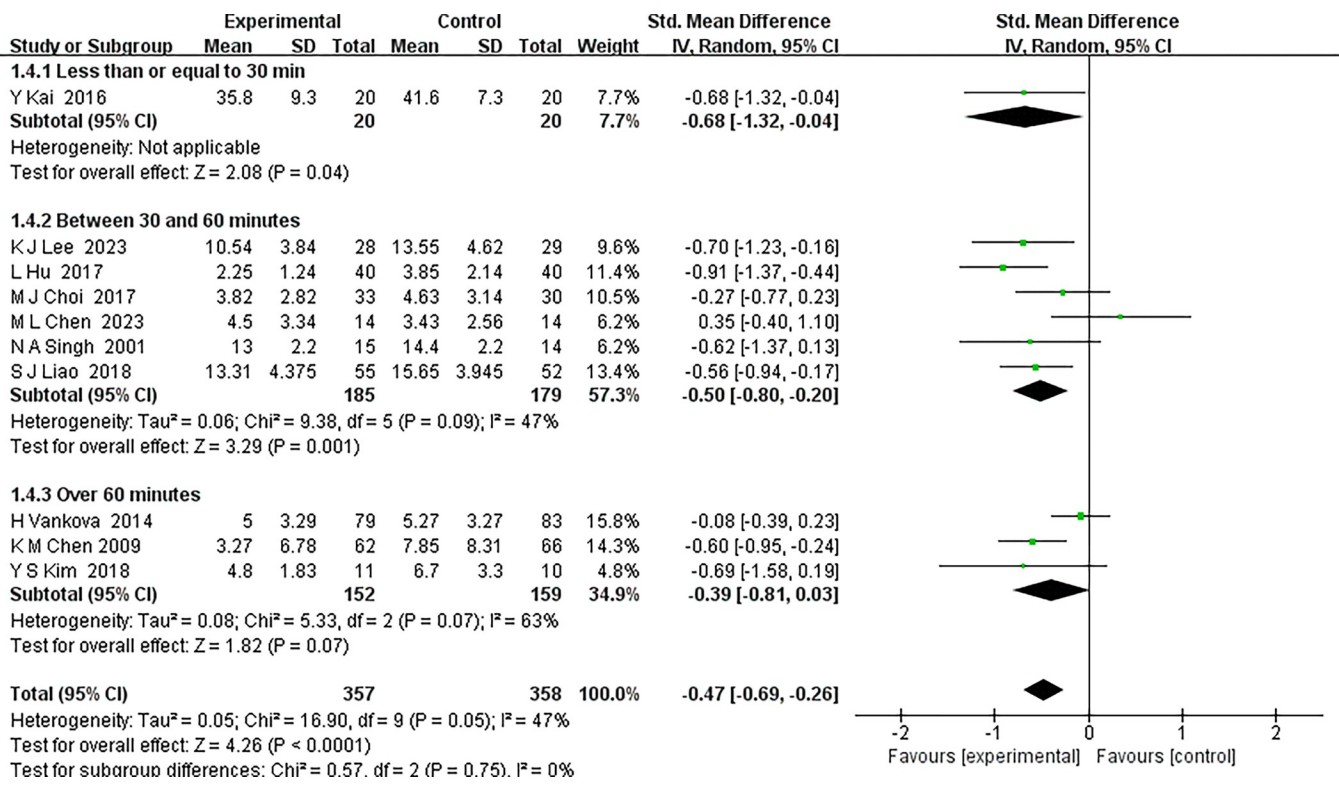

**Fig 8. Forest plot of subgroup analyses of exercise session lengths.**

duration exercise holds substantial efficacy. This underscores the therapeutic potential of moderate-intensity exercise in mitigating depressive symptoms.

3. Long-duration exercise subgroup (>60 minutes): Incorporating three trials, this subgroup exhibited a more modest improvement, with an SMD of -0.39 (95% CI: -0.81 to 0.03). While the trend was towards improvement, the narrow confidence intervals point to potentially lower consistency for longer exercise durations.

Collectively, these outcomes propose that an exercise duration of 30 to 60 minutes might be most beneficial for middle-aged and older individuals with depression, optimizing the improvement in depressive symptoms. Nevertheless, these conclusions must be approached with caution and warrant further validation, especially considering the limited number of studies in certain subgroups, like the short-duration exercise group. The overall high heterogeneity observed in this meta-analysis ($I^2 = 47\%$) may reflect variations in study design, participant demographics, or the specific modalities of exercise interventions across the studies. Future research endeavors should aim to control for these potential sources of heterogeneity. This would enable more precise recommendations regarding the optimal duration of exercise for symptom alleviation in middle-aged and older adults with depression, thereby contributing to the refinement of exercise-based therapeutic interventions for this demographic.

*3.4.4.4 Duration of the exercise programmes.* In this subgroup analysis, we meticulously consolidated studies encompassing various durations of exercise interventions, stratifying them into three distinct subgroups based on intervention length: ≤8 weeks, >8 weeks and ≤16 weeks, and >16 weeks. Remarkably, each exercise intervention duration was linked with significant reductions in depressive symptoms. Illustrated in Fig 9, the analysis revealed:

1. Short-term interventions (≤8 weeks): Displayed a pronounced effect, with a combined Standardized Mean Difference (SMD) of -0.69 (95% Confidence Interval [CI] [-1.10, -0.28]), indicating a statistically significant reduction in depressive symptoms.

2. Medium-term interventions (>8 weeks to ≤16 weeks): Demonstrated a beneficial impact with a combined SMD of -0.34 (95% CI [-0.63, -0.04]), further endorsing the positive influence of exercise on depression.

3. Long-term interventions (>16 weeks): Also showed sustained benefits, with a combined SMD of -0.40 (95% CI [-0.75, -0.06]), supporting the long-term efficacy of exercise interventions.

Collectively, the cumulative SMD for exercise intervention stood at -0.41 (95% CI [-0.60, -0.22]), signifying a significant therapeutic effect of exercise in alleviating depressive symptoms in middle-aged and older patients. It is noteworthy, however, that the analysis of heterogeneity revealed the greatest variation in the subgroup with an intervention duration of 8 to 16 weeks ($I^2 = 58\%$). Although the intervention effect was consistently positive across all timeframes, this heterogeneity may suggest that factors such as varying study designs, differences in intervention intensity, and the baseline characteristics of participants could contribute to the variability in outcomes. These findings underscore the potential of exercise interventions of varying durations to significantly reduce depressive symptoms in the targeted demographic. Nevertheless, the observed heterogeneity invites further investigation into the specific attributes of each intervention category to optimize their effectiveness and tailor them more precisely to individual patient needs.

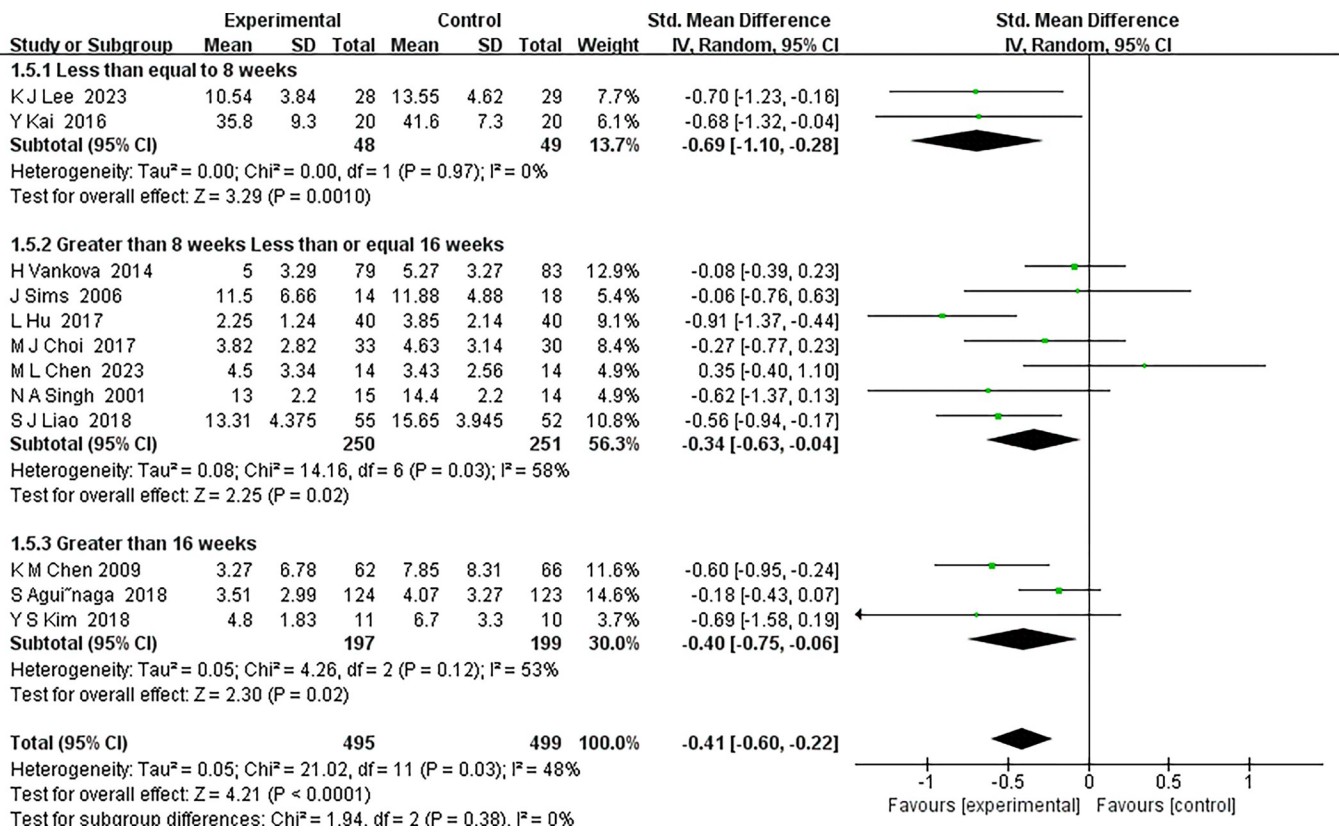

**Fig 9. Forest plot of subgroup analyses of duration of the exercise programmes.**

## 4 Discussion

We executed an exhaustive search across four distinct databases. Our analyses revealed a moderate effect of exercise on depressive symptoms in middle-aged and older adults. Delving into the subgroup analyses of individual versus group exercises, we observed a marginally higher standardised mean difference in individual exercise interventions. This finding hints at a potentially stronger impact of individual exercises in alleviating depressive symptoms in the target demographic. However, subgroup difference tests did not yield statistically significant variances. Consequently, it remains inconclusive whether individual interventions possess a statistical edge over group interventions, despite the marginally elevated overall effect score of the former. This ambiguity may stem from variables like sample sizes, study designs, and the duration and intensity of interventions. Thus, our findings suggest that both individual and group exercise formats are efficacious in managing depression among middle-aged and older individuals, with no substantial difference in their effectiveness.

Our investigation into exercise interventions for ameliorating depression in middle-aged and older adults indicates differential efficacies among various exercise types. Notably, the Gentle Balance series emerged as the most effective, demonstrating the most substantial impact on depressive symptoms. The Interactive Technology Fitness series, albeit based on a smaller dataset, also manifested significant improvements. Meanwhile, aerobic exercise and resistance training exhibited more moderate effects, yet still contributed evidence of potential symptom alleviation. Crucially, the low heterogeneity observed across studies ($I^2 = 11\%$) bolsters the reliability of our meta-analysis outcomes. These findings underscore the necessity of

nuanced subgroup analyses to refine and optimize exercise programs for the middle-aged and elderly suffering from depression.

From a physiological standpoint, independent of the exercise intervention's nature, physical activity enhances neurotrophic factor levels, stimulates relevant adipocytokine production, regulates neurotransmitter expression, augments mitochondrial functionality and melatonin secretion, curbs inflammatory pathways, and influences microRNA expression [40]. The Gentle Balance series, with its emphasis on the mind-body nexus, achieves harmony through continuous posture and breath adjustments. While aerobic and strength training offer mental health benefits, their suitability for middle-aged and older adults might be constrained by age-related physical limitations. Conversely, the Gentle Balance's milder movements align well with the physical characteristics of this demographic, averting the potential exacerbation of depressive symptoms due to overexertion. Hence, for this population, exercise should be a therapeutic intervention rather than an additional burden.

In our exploration of optimal exercise session durations for mitigating depression in middle-aged and older adults, subgroup analyses reveal a preferential impact for sessions lasting between 30 to 60 minutes. This duration appears to maximize benefits in alleviating depressive symptoms within this demographic. However, the conclusions drawn here should be approached with caution due to the limited number of studies, especially concerning shorter-duration exercise subgroups. These preliminary findings necessitate further validation in future research endeavors. The present meta-analysis registered a high overall heterogeneity ($I^2 = 47\%$), likely reflective of the varied study designs, participant demographics, and exercise intervention types. Future research should aim to control these variables to more accurately pinpoint the most effective exercise duration for depression relief in middle-aged and older adults. Notably, in our subgroup analysis focused on exercise intervention durations, all examined durations correlated with significant reductions in depressive symptoms. The most pronounced effect was observed in short-term interventions, followed by medium- and long-term regimens. These insights are pivotal for the development of tailored exercise programs designed to address depressive symptoms in the middle-aged and older population.

Our investigation contributes to the growing body of evidence supporting the efficacy of exercise interventions in mitigating depressive symptoms among middle-aged and older adults. This aligns with the seminal findings of Blumenthal et al. (1999), who demonstrated a significant reduction in depressive symptoms following exercise interventions in this demographic [41]. However, our analysis revealed a comparatively modest effect size in contrast to the results presented by Rhyner et al. (2016) [42]. This variance can likely be attributed to the diminished intensity and frequency of the exercise regimens employed in our study, coupled with our focus on individuals exhibiting primarily mild to moderate depression—distinct from Rhyner et al.'s inclusion of participants across a broader spectrum of depression severity. Furthermore, our utilization of an alternative tool for assessing depressive symptoms might have contributed to the observed discrepancies, underscoring the necessity for future research to adopt standardized measurement approaches. Such studies should also explore the impact of varying exercise intervention intensities on depressive symptom amelioration, to refine and optimize therapeutic strategies.

This investigation delineates the efficacy of exercise interventions in mitigating depressive symptoms among middle-aged and older adults, laying a robust groundwork for the application of these insights in both clinical practice and public health policy. It advocates for an initial assessment by healthcare professionals to discern the precise requirements of their patients, thereby facilitating the customization of exercise regimens, a critical step towards the integration of such interventions into clinical settings. Furthermore, it urges policy makers to embed exercise interventions within public health initiatives, for instance, through the provision of community exercise amenities and the orchestration of ongoing health promotion

endeavors. Emphasis is placed on the strategic allocation of resources to guarantee the broad accessibility and practicality of these interventions, particularly in settings constrained by limited resources. To bolster the recommendations for practice and policy, engagement with key stakeholders—encompassing healthcare practitioners, community figureheads, and policy architects—is imperative to advocate for the significance of physical activity interventions and to champion their expansive adoption. Moreover, in response to the evolving needs of the middle-aged and elderly demographic, it is paramount to persistently assess and refine the effectiveness of these interventions through continuous evaluation and adaptation, ensuring their alignment with the dynamic health landscape.

## 5 Conclusion

The findings gleaned from this investigation illuminate significant insights. In unraveling the impact of interventions, our study reveals that exercise exerts a discernible influence on alleviating depression symptoms in middle-aged and older individuals. The efficacy of this intervention hinges upon several pivotal factors, such as the mode of exercise (individual or group), the nature of the exercise regimen, the temporal aspect of singular exercise sessions, and the overall duration of engagement. Among the diverse exercise interventions scrutinized, gentle balance exercises emerge as the most potent in ameliorating depression among middle-aged and older individuals. Delving into the subgroup analysis based on individual or group dynamics, it is discerned that solitary exercise outshines its group counterpart in fostering a positive impact on depression in this demographic. Furthermore, the temporal dimension of individual exercise underscores the optimal effectiveness of moderate sessions lasting 30–60 minutes. In the realm of exercise intervention durations, our findings underscore the supremacy of short-term interventions, followed by a commendable efficacy exhibited by moderate and long-term interventions. These nuanced insights emphasize the multifaceted nature of the relationship between exercise and depression in middle-aged and older individuals, paving the way for a more nuanced and tailored approach to intervention strategies.

While our investigation sheds light on the beneficial impacts of exercise interventions, it also underscores the necessity for further research in several domains. Initially, a profound inquiry into the precise mechanisms through which exercise interventions exert their influence is paramount for the crafting of more efficacious treatment strategies. Although our analysis pinpointed specific protocols with beneficial outcomes, deciphering the exact physiological and psychological pathways affected by these interventions will significantly enhance their optimization. Moreover, the exploration of exercise interventions' enduring effects stands as a crucial avenue for forthcoming studies. The body of existing research predominantly scrutinizes short-term outcomes, leaving a gap in our understanding of the long-term benefits. Longitudinal studies focusing on the sustained application of exercise interventions could elucidate their capacity to mitigate depressive symptoms among middle-aged and older individuals, offering a more comprehensive view of their therapeutic potential. Lastly, investigating the variability in response to exercise interventions across different demographics presents a vital research trajectory. Divergent responses among middle-aged and older adults, attributable to variations in gender, cultural background, and socioeconomic status, may reveal insights into how exercise interventions can be tailored to meet the unique needs of each subgroup. Unraveling these nuances is instrumental in devising personalized, more effective treatment modalities.

## Supporting information

**S1 Checklist. Human participants research checklist.**
(DOCX)

**S2 Checklist. PRISMA 2020 checklist.**
(DOCX)

**S1 File.**
(DOCX)

**S2 File.**
(PDF)

## Author Contributions

**Conceptualization:** Jia-Yun Liu, Kai-Tuo Zhu, Gui-Quan Huo.

**Data curation:** Zheng Zhang, Jia-Yun Liu, Kai-Tuo Zhu, Gui-Quan Huo.

**Formal analysis:** Zheng Zhang, Jia-Yun Liu, Kai-Tuo Zhu.

**Writing – original draft:** Zheng Zhang.

**Writing – review & editing:** Zheng Zhang.

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
