## [Decision Letter · Decision Letter 0]

27 Mar 2024

PONE-D-24-06785A META analysis and systematic review of the effects of exercise interventions on middle-aged and elderly patients with depressionPLOS ONE

Dear Dr. ZHANG,

Thank you for submitting your manuscript to PLOS ONE. After careful consideration, we feel that it has merit but does not fully meet PLOS ONE’s publication criteria as it currently stands. Therefore, we invite you to submit a revised version of the manuscript that addresses the points raised during the review process.

We look forward to receiving your revised manuscript.

Kind regards,

Zhiwen Luo

Academic Editor

PLOS ONE

3. Please amend the manuscript submission data (via Edit Submission) to include authors Jia-Yun Liu, Kai-Tuo Zhu, and Gui-Quan Huo.

4. Please remove your figure 1 from within your manuscript file, leaving only the individual TIFF/EPS image files, uploaded separately. These will be automatically included in the reviewers’ PDF.

Additional Editor Comments:

The reviewers made a series of comments on the manuscript, and the authors need to make point-to-point revisions.

Reviewers' comments:

Reviewer's Responses to Questions

**Comments to the Author**

1. Is the manuscript technically sound, and do the data support the conclusions?

Reviewer #1: Yes

Reviewer #2: Yes

2. Has the statistical analysis been performed appropriately and rigorously? 

Reviewer #1: Yes

Reviewer #2: Yes

3. Have the authors made all data underlying the findings in their manuscript fully available?

Reviewer #1: Yes

Reviewer #2: Yes

4. Is the manuscript presented in an intelligible fashion and written in standard English?

Reviewer #1: Yes

Reviewer #2: Yes

5. Review Comments to the Author

Reviewer #1: The study "A Meta-Analysis and Systematic Review of the Effects of Exercise Interventions on Middle-Aged and Elderly Patients with Depression" provides a pioneering exploration of this area from the perspective of meta-analysis, further reinforcing relevant conclusions. Overall, the authors present a manuscript of significant relevance to the scientific community. However, before publication, some modifications are necessary.

Comments:

1. For the quality assessment of each study, authors should consider further visual representations, such as graphs, to enhance clarity and efficiency.

2. There are various methods to assess publication bias; however, the rationale behind the chosen method in this article should be clarified.

3. While mentioning the necessity of future research in the conclusion, it is advisable to delve deeper into discussing the directions of future studies, such as exploring specific intervention mechanisms, long-term effects, and intergroup differences.

4. Given the observed heterogeneity in the study, it is recommended to provide a more in-depth discussion on the impact and management of this heterogeneity to enhance readers' understanding and trust in the study results.

5. In the discussion section, further exploration of the consistency or discrepancies with existing literature and the potential reasons behind these differences is warranted.

6. Further discussion on the implementation and feasibility of intervention measures, as well as how to translate these research findings into clinical practice or public health policies, would be beneficial.

7. The use of meta-analysis methods is widespread. The following articles are worthy of study and citation: (PMID: 36740666; 35960093; 35776991).

8. Please thoroughly check for spelling and punctuation errors in the manuscript. This manuscript requires appropriate revisions.

9. Some grammatical errors were noted in the discussion section, which should be corrected through proofreading.

Reviewer #2: The research study conducted a meta-analysis and systematic review to explore the effects of exercise interventions on middle-aged and elderly patients with depression. The investigators extensively searched multiple databases and identified 12 relevant papers involving 994 subjects. The meta-analysis indicated a significant overall effect size of exercise interventions on depressive symptoms in this demographic. However, some heterogeneity among studies was observed.

Q1 The introduction should provide more background information on the prevalence and impact of depression in middle-aged and older adults.

Specify the inclusion and exclusion criteria used during the screening process for selecting papers.

Q2 Describe the characteristics of the included subjects (e.g., age range, gender distribution) in more detail.

Q3 Clarify the primary outcome measures used to assess depressive symptoms in the selected studies.

Q4 Provide a flow diagram illustrating the study selection process, as recommended by PRISMA guidelines.

Q5 Discuss the potential sources of heterogeneity among the included studies in the discussion section.

Q6 Consider conducting subgroup analyses based on exercise intensity, duration, and type.

Address the study's limitations, including potential publication bias and the small number of included papers.

Q7 Elaborate on the implications of the findings and how they contribute to the existing literature.

Q8 Provide recommendations for clinicians or policymakers based on the study's results.

Include a brief description of the statistical methods employed in the meta-analysis.

Specify the software used for data analysis and provide details on the statistical models utilized.

Q9 Consider including a forest plot to present the effect sizes of individual studies visually.

Provide a clear definition of "gentle and balanced exercise series" and explain why it is considered particularly efficacious.

Q10 Justify the choice of moderate exercise sessions lasting 30 to 60 minutes as optimal.

Discuss any potential adverse effects or risks associated with exercise interventions in this population.

Q11 Elaborate on the implications of short-term, medium-term, and long-term exercise interventions regarding effectiveness.

Q12 Discuss possible mechanisms underlying the positive effect of exercise on depressive symptoms.

Q13 Offer suggestions for future research directions based on the study's limitations and gaps in knowledge.

Q14 Proofread the manuscript for grammatical errors and ensure consistency in writing style.

6. PLOS authors have the option to publish the peer review history of their article (what does this mean?). If published, this will include your full peer review and any attached files.

Reviewer #1: No

Reviewer #2: No

---

## [Author Response · Author response to Decision Letter 0]

8 Apr 2024

Dear editor, I have made the changes as per your request, thanks very much for your suggestion, regarding the reviewer's response, I have uploaded the attachment, thanks again!

---

## [Decision Letter · Decision Letter 1]

29 Apr 2024

A META analysis and systematic review of the effects of exercise interventions on middle-aged and elderly patients with depression

PONE-D-24-06785R1

Dear Dr. ZHANG,

We’re pleased to inform you that your manuscript has been judged scientifically suitable for publication and will be formally accepted for publication once it meets all outstanding technical requirements.

Kind regards,

Zhiwen Luo

Academic Editor

PLOS ONE

Additional Editor Comments (optional):

Reviewers' comments:

Reviewer's Responses to Questions

**Comments to the Author**

1. If the authors have adequately addressed your comments raised in a previous round of review and you feel that this manuscript is now acceptable for publication, you may indicate that here to bypass the “Comments to the Author” section, enter your conflict of interest statement in the “Confidential to Editor” section, and submit your "Accept" recommendation.

Reviewer #1: All comments have been addressed

Reviewer #2: All comments have been addressed

2. Is the manuscript technically sound, and do the data support the conclusions?

Reviewer #1: Yes

Reviewer #2: Yes

3. Has the statistical analysis been performed appropriately and rigorously? 

Reviewer #1: Yes

Reviewer #2: Yes

4. Have the authors made all data underlying the findings in their manuscript fully available?

Reviewer #1: Yes

Reviewer #2: Yes

5. Is the manuscript presented in an intelligible fashion and written in standard English?

Reviewer #1: Yes

Reviewer #2: Yes

6. Review Comments to the Author

Reviewer #1: The study "A Meta-Analysis and Systematic Review of the Effects of Exercise Interventions on Middle-Aged and Elderly Patients with Depression" provides a pioneering exploration of this area from the perspective of meta-analysis, further reinforcing relevant conclusions. Overall, the authors present a manuscript of significant relevance to the scientific community.

The author has made corresponding modifications according to the requirements. Agreed to publish.

Reviewer #2: In response to the comments I made, the author has already made satisfied adjustments, and I recommended that it be published.

7. PLOS authors have the option to publish the peer review history of their article (what does this mean?). If published, this will include your full peer review and any attached files.

Reviewer #1: No

Reviewer #2: No

---

## [Editor Report · Acceptance letter]

31 Aug 2024

PONE-D-24-06785R1 

PLOS ONE

Dear Dr. Zhang, 

I'm pleased to inform you that your manuscript has been deemed suitable for publication in PLOS ONE. Congratulations! Your manuscript is now being handed over to our production team.

Kind regards, 

on behalf of

Dr. Zhiwen Luo 

Academic Editor

PLOS ONE